# ANTI-REFERENCE: UNIVERSAL AND IMMEDIATE DEFENSE AGAINST REFERENCE-BASED GENERATION

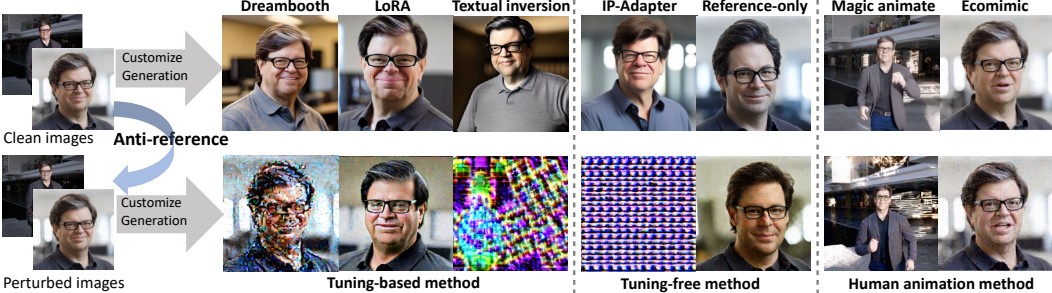

Figure 1: Malicious attackers can collect users' images as reference images and use diffusion models to achieve malicious purposes. Our system, called Anti-reference, applies imperceptible perturbations to user-uploaded images before they are published, resulting in noticeable artifacts in images or videos generated by reference-based methods and fine-tuning approaches. This makes it easy to recognize them as AI-generated, thus protecting the images.

## ABSTRACT

Diffusion models have revolutionized generative modeling with their exceptional ability to produce high-fidelity images. However, misuse of such potent tools can lead to the creation of fake news or disturbing content targeting individuals, resulting in significant social harm. In this paper, we introduce Anti-Reference, a novel method that protects images from the threats posed by reference-based generation techniques by adding imperceptible adversarial noise to the images. We propose a unified loss function that enables joint attacks on fine-tuning-based customization methods, non-fine-tuning customization methods, and human-centric driving methods. Based on this loss, we train a Adversarial Noise Encoder to predict the noise or directly optimize the noise using the PGD method. Our method shows certain transfer attack capabilities, effectively challenging both gray-box models and some commercial APIs. Extensive experiments validate the performance of Anti-Reference, establishing a new benchmark in image security.

## 1 INTRODUCTION

Customized diffusion models can be divided into methods that require training, Ruiz et al. (2023); Hu et al. (2021); Gal et al. (2022); Kumari et al. (2023) and those that do not, such as IP-Adapter (Ye et al., 2023a), Instant-ID (Wang et al., 2024b). Reference-based methods are widely used in customized image and video generation, especially in human-centered video generation, including portrait video creation methods(Tian et al., 2024; Chen et al., 2024; He et al., 2024; Xie et al., 2024), and human animation (Xu et al., 2024; Hu, 2024) , which have attracted significant attention due to their practical value in creating digital human avatars and enhancing film production.

Reference-based methods that require no training offer high convenience and efficiency, but when misused, they can have severe negative social impacts, such as creating fake news or pornographic images targeting individual victims. Existing studies use encoder attack (Salman et al., 2023) and diffusion attack (Van Le et al., 2023; Liang et al., 2023) to protect images from the threats posed by methods requiring fine-tuning, using PGD (Madry, 2017) optimization to generate adversarial noise, but this approach requires several minutes to protect a single image, severely limiting its practical application. Moreover, these methods are largely ineffective against non-trainable Reference-based generation methods. Therefore, developing an efficient method to protect personal images from the threats of Reference-based generation has become an urgent priority.

Reference-based methods provide additional conditions through a Reference Image to enable customized generation. These methods can be divided into two types based on their implementation: one type embeds Reference features in the cross-attention layer of the denoising network using an adapter, such as IP-Adapter (Ye et al., 2023a); the other type embeds reference features in the self-attention layer of the denoising network using ReferenceNet. The approach of ReferenceNet is widely used for image customization generation (Team, 2023; Zhang et al., 2024b;c), Image2Video (Chen et al., 2023; Zhang et al., 2023), and face animation generation (Tian et al., 2024; Chen et al., 2024; He et al., 2024; Xie et al., 2024), and body-driven tasks (Xu et al., 2024; Hu, 2024). However, due to the variety of existing Reference-based generation methods, attacking a specific method has limited practical significance, as attackers can easily switch methods to bypass protection. Therefore, the motivation of this paper is to propose a universal adversarial noise generation method to address the threats posed by mainstream Reference-based methods.

In practical image protection scenarios, protection methods need to address several challenges. Firstly, universality is a key challenge. Since Reference-based methods have many different implementations, and models trained on different datasets have different feature spaces, the same attack strategy may have very different effects on different models. Secondly, efficiency is also crucial. Existing methods like Anti-DreamBooth (Van Le et al., 2023) , which use PGD optimization, usually require hundreds of steps and significant time, severely limiting their feasibility for real-time applications. Finally, black-box or gray-box transferability and robustness are also central challenges. In practical applications, the structures and parameters of proprietary APIs like EMO (Tian et al., 2024) , Animate anyone (Hu, 2024) are not accessible, so attack methods must have good gray-box transferability. Additionally, the generated adversarial noise also needs to be robust enough to withstand common data augmentation operations and preprocessing steps.

To address these challenges, this paper presents Anti-Reference, the first to protect images from the threats posed by mainstream reference-based methods and tuning-based customization methods through the forward process. We propose a Noise Encoder based on the ViT (Dosovitskiy, 2020) architecture, which predicts adversarial noise of the same size as the original image and overlays it to form a protected image. To achieve a universal attack on methods requiring fine-tuning and those that do not, we designed a unified loss function, using a weighted strategy to achieve joint attack effects across multiple tasks, and by limiting the noise range and regularization loss to ensure the invisibility of the noise. To enhance the robustness of adversarial noise, we also introduced some data augmentation techniques to ensure that the adversarial noise can withstand various data enhancements and preprocessing operations. As the model structures and weights of proprietary APIs are not accessible, directly attacking these models is usually not feasible. To overcome this hurdle, we created white-box proxy models that mimic the structure and behavior of these proprietary models, and we successfully implemented attacks on these proxy models, thereby achieving gray-box transferability attacks. Specifically, our adversarial samples have successfully transferred to closed-source APIs (such as Animate Anyone (Hu, 2024) and EMO (Tian et al., 2024)). Extensive experimental results demonstrate that Anti-Reference is highly effective in protecting images from potential security threats posed by reference-based generation methods and fine-tuning-based approaches.

We summarize our main contributions as follows:

- We introduce a universal method for attacking customized diffusion models for the first time, which is effective against both mainstream reference-based generation methods and those requiring fine-tuning.
- We introduce an Adversarial Noise Encoder that executes attacks without the need for traditional PGD optimization, significantly reducing computational time and enhancing suitability for real-time applications.
- We have designed transferable adversarial samples that enable gray-box attacks on commercial APIs using white-box proxy models. These samples are robust, showing strong resistance to common image transformations.

## 2 RELATED WORK

### 2.1 CUSTMIZED DIFFUSION MODEL.

Diffusion probability models Song et al. (2020); Ho et al. (2020) represent a class of advanced generative models that reconstruct original data from pure Gaussian noise by learning noise distributions

at different levels. These models excel in handling complex data distributions and have marked significant accomplishments across various fields such as image synthesis Rombach et al. (2021); Peebles & Xie (2023), image editing Brooks et al. (2023); Hertz et al. (2022), video generation Wu et al. (2022); Hu (2024), and 3D content creation Poole et al. (2022). A prominent example is Stable Diffusion Rombach et al. (2021), which utilizes a UNet architecture to iteratively produce images, demonstrating robust text-to-image capabilities after extensive training on large text-image datasets. DreamBooth Ruiz et al. (2023), Custom diffusion Kumari et al. (2023) and Textual Inversion Gal et al. (2022), adopt transfer learning to text-to-image diffusion models via either fine-tuning all the parameters, partial parameters , or introducing and optimizing a word vector for the new concept. LoRA (Low-Rank Adaptation) Hu et al. (2021) is a popular and lightweight training technique that significantly reduces the number of trainable parameters and is widely used for personalized or task-specific image generation.

## 2.2 Reference-based Generation

In addition to the aforementioned fine-tuning methods, finetuning-free customized generation methods can capture concepts from a single image and are widely used for tasks such as customized generation (Ye et al., 2023a; Mao et al., 2024; Zhang et al., 2024a), identity consistency maintenance (Wang et al., 2024b; Li et al., 2024), face-driven Tian et al. (2024); Chen et al. (2024); Xie et al. (2024), and body-driven tasks Xu et al. (2024); Hu (2024). These methods can be roughly categorized into the Adapter approach and the ReferenceNet approach based on how the reference image features are utilized. In the Adapter approach, the reference image is first processed by a pre-trained image feature extractor, typically CLIP (Radford et al., 2021) image encoder or ArcFace Deng et al. (2019), and then an adapter structure generates visual tokens applied to the cross-attention layers of the U-Net. The ReferenceNet approach emphasizes the effectiveness of integrating reference image features into the self-attention layers of LDM U-Nets, enabling customized generation while preserving appearance context. Image-to-video technology Chen et al. (2023); Zhang et al. (2023) uses ReferenceNet to maintain consistency between the generated results and the reference image. Magic Animate Xu et al. (2024) and Animate Anyone Hu (2024) combine ReferenceNet with pose control and temporal modules to achieve body-driven generation. EMO Tian et al. (2024), Ecomimic Chen et al. (2024), and X-Portrait Xie et al. (2024), among other talking-face methods, maintain identity consistency using ReferenceNet, generating fake videos from just a single photo. The misuse of Reference-based Generation methods can have severe consequences, making it urgent to protect images from the threats posed by such methods.

## 2.3 Protective Perturbation against Diffusion.

Protecting the security of personal images is of great significance Dong et al. (2023); Qiao et al. (2024); Dai et al. (2024) . To protect personal images such as faces and artwork from potential infringement when used for fine-tuning Stable Diffusion, recent research aims to disrupt the fine-tuning process by adding imperceptible protective noise to these images. Several methods have been developed to achieve this goal: Glaze (Shan et al., 2023) focuses on preventing artists' work from being used for specific style mimicry in Stable Diffusion. It optimizes the distance between the original image and the target image at the feature level, causing Stable Diffusion to learn the wrong artistic style. AdvDM (Liang et al., 2023) proposes a direct adversarial attack on Stable Diffusion by maximizing the Mean Squared Error loss during the optimization process. This approach uses adversarial noise to protect personal images. Anti-DreamBooth (Van Le et al., 2023) incorporates the DreamBooth fine-tuning process of Stable Diffusion into its consideration. It designs a bi-level min-max optimization process to generate protective perturbations. Additionally,other research efforts (Wang et al., 2024a; Ye et al., 2023b; Zheng et al., 2023) have explored generating protective noise for images using similar adversarial perturbation methods.

The previously mentioned methods utilize adversarial noise to influence the fine-tuning process, preventing models from learning from tampered images. These techniques effectively target models that require fine-tuning. However, reference-based generation methods do not rely on fine-tuning but directly generate images from existing data, making these adversarial protections ineffective against them. Effective protection against reference-based generation attacks requires new strategies that can directly intervene in the image retrieval and matching mechanisms. Effective protection against reference-based generative attacks requires the development of new strategies.

Figure 2: Illustration of Anti-reference. To defend against customized generation attacks, we introduce a loss function that guides the training of a noise encoder to produce adversarial perturbations, effectively protecting source images.

## 3 PROBLEM DEFINITION

Considering the practical implications of image infringement based on Stable Diffusion, it is essential to define the threat model in real-world scenarios. We consider two participants involved in fine-tuning Stable Diffusion using images: the "User" Alice and the "Photo Thief" Bob. Photo Thief Bob illicitly uses reference-based methods to exploit others' photos for customized content, while User Alice, wishing to safeguard her images on social media, adds adversarial noise to disrupt Bob's methods, aiming to induce severe artifacts in the generated content. Specifically, we explain the workflow of the two parties as follows:

**User Alice:** Alice aims to protect her images from exploitation by Stable Diffusion by applying nearly imperceptible protective perturbations, while minimizing alterations to the original images. Her main challenge is the uncertainty of which methods Photo Thief Bob will use to fine-tune these protected images. She also needs to ensure that these protection measures remain effective even when the images undergo natural transformations such as cropping, compression, and blurring during dissemination.

**Photo Thief Bob:** Bob downloads Alice's photos and uses customized generation methods to create inappropriate content. Bob can choose any mainstream fine-tuning method, including but not limited to direct fine-tuning, LoRA, Textual Inversion, DreamBooth, or Custom Diffusion.

The goal of this work is to add imperceptible adversarial noise to images, formalized as $I_{adv} = I + noise$, where $I$ and $I_{adv}$ represent the original and protected images, respectively. These images serve as inputs to customization methods, and the outputs $\text{Gen}(I)$ and $\text{Gen}(I_{adv})$ are compared. If $\text{Gen}(I_{adv})$ exhibits significant distortion, the protection is considered successful. We achieve this by solving the following optimization problem:

$$\max_{x_{adv} \in M} d(\text{Gen}(I), \text{Gen}(I_{adv})) \text{ subject to } d'(I, I_{adv}) \leq \delta, \tag{1}$$

where $M$ indicates the natural image manifold, $d$ and $d'$ denote image distance functions, and $\delta$ represents the fidelity budget. Through this optimization process, we aim to effectively safeguard images from unauthorized editing and translation while maintaining their fidelity.

## 4 METHOD

In Sec. 4.1, we present the overall framework, followed by details of the Noise Encoder (Sec. 4.2) and the loss function (Sec. 4.3). Sec. 4.4 describes PGD joint optimization, and Sec. 4.5 explains white-box proxy construction for gray-box attacks.

### 4.1 OVERALL METHOD

This section introduces the overall framework of the Anti-Reference method, as shown in Fig. 2. Our method consists of several key components: the Noise Encoder, a set of conditional modules, the Denoising Unet, and a differentiable data augmentation module. The Noise Encoder adds adversarial noise to the image, forming the protected image $I_{adv}$. The set of Reference Modules is a group of conditional control modules that serve as the target models for the attack.

To protect images from the threats posed by tuning-free customization generation methods and driving methods, we selected the pre-trained ReferenceNet from Magic Animate and Ecomimic, as well as the Stable Diffusion Unet, as the target models for attacking the ReferenceNet route. Additionally, we chose the IP-Adapter as the target model for the Adapter route. The Denoising Unet utilizes the pre-trained Stable Diffusion 1.5 Unet, as it is the most commonly used base model for various customization generation methods. The protected image $I_{adv}$ is fed into two components: the set of conditional modules and the Denoising Unet, where losses are calculated separately. To enhance the robustness of the adversarial noise against real-world scenarios, we propose a differentiable data augmentation module, which applies common data augmentations to $I_{adv}$.

## 4.2 ADVERSARIAL NOISE ENCODER

We propose an adversarial noise encoder (ANE) based on the Vision Transformer (ViT) Dosovitskiy (2020) to efficiently generate adversarial noise in the pixel space, protecting images from threats posed by generative models. The design of the encoder incorporates the following key technical details: ANE adopts the ViT architecture with 12 Transformer layers, a hidden size of 384, and 6 attention heads. The input image is divided into 8×8 patches, making it well-suited for detailed feature extraction and adversarial noise predict. The sequence is processed through multiple layers of self-attention and feedforward network modules, resulting in feature vectors. ANE directly generates adversarial noise in the pixel domain instead of relying on latent space.

To enhance robustness, we adopt adversarial training during the training process, including random cropping and scaling, JPEG compression, Gaussian noise, and color transformations. These data augmentation techniques improve the stability of the noise in real-world scenarios, ensuring its effectiveness even after preprocessing or compression. In the training process, to prevent noise from falling into local optima, noise amplitude is regulated through gradient constraints. The model is trained at a resolution of 512×512, maintaining alignment with the common settings of the target generative methods, thereby ensuring compatibility and effectiveness across various generation tasks.

We found that if the conditional model and the denoising Unet shown in Fig. 2 are fixed, ANE tends to generate simple adversarial noise patterns (such as targeting specific vulnerabilities) rather than comprehensively robust noise. This "speculative" behavior may weaken the generator's generalization ability. To address this, we employ a phased training approach to enhance ANE's adaptability. In the first phase: the denoising Unet and three kinds of conditional models (IP-Adapter Ye et al. (2023a) and 3 ReferenceNet Chen et al. (2024); Team (2023); Xu et al. (2024) ) are fixed, and ANE is trained to identify effective attack strategies quickly. In the second phase: we randomly perturb the impact weights of the conditional models and switch between different customized models every 1000 steps during training, including replacing the Unet and attaching stylized LoRA Hu et al. (2021) plugins. We obtain these models from the Civitai civ community.

## 4.3 LOSS FUNCTION

**Diffusion Adversarial loss.** In the context of diffusion, in Formula equation 1, which involves maximizing the difference between two images, is transformed into maximizing the difference in noise prediction. Anti-Dreambooth Van Le et al. (2023) was the first to adopt this approach, which was then utilized by subsequent methods Wang et al. (2024a); Ye et al. (2023b); Zheng et al. (2023). This means that we aim for the noise predicted by the model, $\epsilon_\theta$, to have the largest possible error compared to the actual noise $\epsilon$, thereby disrupting the model's denoising capability. The specific loss function can be defined as:

$$L_{\text{adv}} = -\mathbb{E}_{x_0, \epsilon \sim \mathcal{N}(0,1), t} \left[ \|\epsilon - \epsilon_\theta(x_t, t)\|^2 \right], \tag{2}$$

where $x_0$ is the original data, $\epsilon$ is noise sampled from a standard normal distribution, $t$ is the time step representing the noise level, $x_t = \sqrt{\bar{\alpha}_t} x_0 + \sqrt{1 - \bar{\alpha}_t} \epsilon$ is the noisy image at time step $t$, $\epsilon_\theta(x_t, t)$ is the noise predicted by the model. This loss function is as same as diffusion training loss, but the objective is completely opposite.

**Conditional Adversarial Loss.** Conditional Adversarial Loss aims to attack reference-based customization generation methods and driving techniques. Specifically, we calculate the adversarial noise prediction loss when adversarial noise images are used as inputs for ReferenceNet or IP-adapter. This loss deviates the noise predicted by the denoising Unet from the ground-truth noise, under specific conditional features provided by either ReferenceNet or the IP-adapter. The conditional

adversarial loss is formulated as follows:

$$L_{\text{con\_adv}} = -\mathbb{E}_{x_0, \epsilon \sim \mathcal{N}(0,1), t, c} \left[ \|\epsilon - \epsilon_\theta(x_t, t, c)\|^2 \right], \tag{3}$$

$c$ represents the features extracted from $I_{adv}$ using ReferenceNet or IP-adapter. These features interfere with the denoising process by injecting signals into the Unet's cross- or self-attention layers.

**Image Regularization Loss.** To make the adversarial noise less perceptible, we calculate the Mean Squared Error (MSE) of the images before and after noise addition as the regularization loss.

$$L_{\text{reg}} = \text{MSE}(I, I_{adv}) \tag{4}$$

**Total Loss.** For joint attacks, a weighted loss formulation is employed to ensure a balanced attack performance across various tasks by balancing the impact across all contributions. The total loss, incorporating adversarial, conditional adversarial, and regularization losses, is defined as follows:

$$L_{\text{total}} = w_{\text{adv}} \cdot L_{\text{adv}} + \sum_i w_{\text{con},i} \cdot L_{\text{con\_adv},i} + w_{\text{reg}} \cdot L_{\text{reg}}, \tag{5}$$

where, $w_{\text{con},i} \cdot L_{\text{con\_adv},i}$ represents the weighted sum of conditional adversarial losses from different conditional modules. Each module $i$ targets different conditional control tasks, and $w_{\text{con},i}$ is the specific weight assigned to the conditional adversarial loss for module $i$. This paper conducts joint training across four conditional modules: IP-Adapter Ye et al. (2023a), Reference-only Team (2023), Magic Animate Xu et al. (2024), and Ecomimic's ReferenceNet Chen et al. (2024). This approach allows for tailored defenses against a range of adversarial manipulations facilitated by different attack modules, ensuring that the influence of each module is properly scaled according to its significance and effectiveness in the overall defense strategy.

## 4.4 PGD JOINT OPTIMIZATION

We introduce our Anti-Reference (PGD) method, where adversarial noise is optimized directly using PGD (Projected Gradient Descent). PGD iteratively perturbs the input image $I$ within a predefined bound, ensuring the noise remains imperceptible while maximizing its impact on the model's predictions. Unlike the Noise Encoder, which generates noise in a single pass, PGD updates the noise iteratively by calculating the gradient of the loss function with respect to the image. At each iteration, the adversarial noise is updated as:

$$I_{adv}^{(k+1)} = \Pi_{I+\epsilon} \left( I_{adv}^{(k)} + \alpha \cdot \text{sign} \left( \nabla_{I_{adv}^{(k)}} L_{\text{total}} \right) \right), \tag{6}$$

where, $I_{adv}^{(k)}$ is the adversarial image at iteration $k$, $\alpha$ is the step size, and $\epsilon$ defines the perturbation bound. The projection $\Pi_{I+\epsilon}$ ensures the noise stays within the allowed limits.

By optimizing both $L_{\text{adv}}$ and $L_{\text{con\_adv}}$, PGD effectively disrupts both the diffusion process and conditional adversarial predictions. Our experiments show that PGD provides strong protection across various reference-based customization methods, with gradually increasing noise impact while preserving image quality. Although the Noise Encoder generates noise faster, PGD's iterative process offers stronger protection across tasks at a higher computational cost, making it ideal for scenarios demanding maximum protection.

## 4.5 GRAY-BOX TRANSFER

This section introduces proxy-based gray-box attacks, a method that generates adversarial samples using a white-box model with a structure similar to the target gray-box model or a closely related latent space. By training DiT to generate adversarial samples on the white-box model, these samples also achieve high attack success rates on the gray-box model. The success of this approach relies on two key factors: 1) structural similarity between the white-box and gray-box models, and 2) shared similarity in their latent spaces. For instance, both Animate Anyone and Magic Animate are based on Stable Diffusion 1.5 and share the same ReferenceNet architecture, with similar datasets used for fine-tuning, resulting in similar latent spaces. Additionally, we successfully attacked the EMO Tian et al. (2024), Animate anyone Hu (2024) and other apps or APIs, as demonstrated in the experiments.

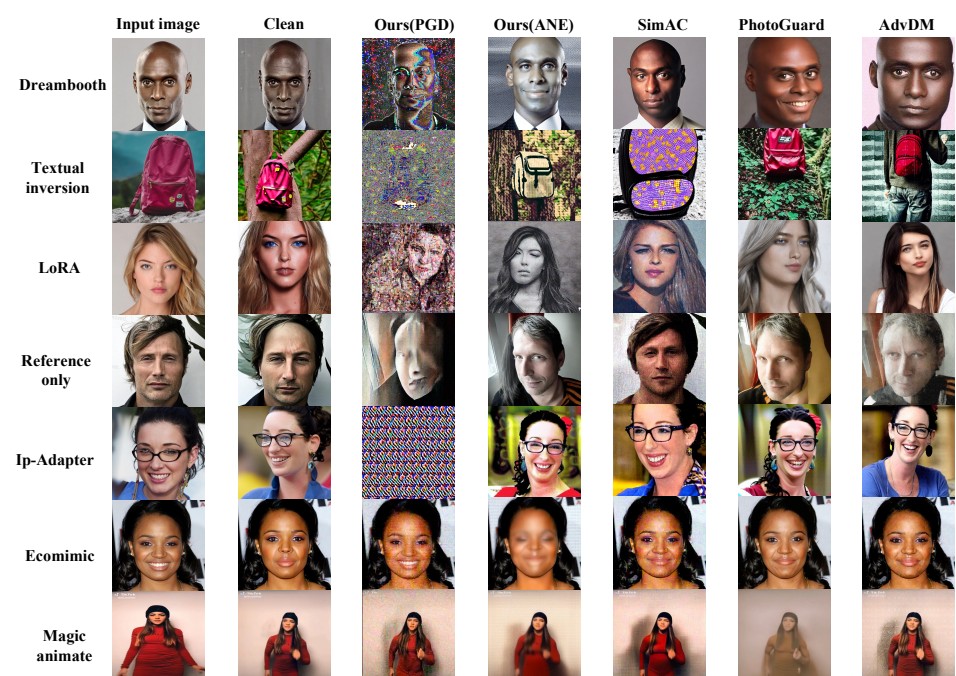

Figure 3: Results of protection methods against customized generation threats. Our approach delivers strong and comprehensive attack performance across scenarios.

# 5 EXPERIMENT

## 5.1 SETUP

**Training data.** This paper aims to achieve general image protection, and therefore, we use 600K natural image-text pairs from the Laion dataset as the training set. To enhance the protection effectiveness for talking face and body-driven tasks, we also include the Celeb-A dataset (200K) and the TikTok dataset (30K) into the training data.

**Experimental details.** We used 4 A100 GPUs to train on 830K image-text pairs for 4 epochs with a batch size of 8, employing a learning rate decay strategy with an initial value of $10^{-3}$. We utilized a pre-trained DiT-S/8 model with the same architecture as ANE for the Noise Encoder. During ANE training, adversarial noise is unrestricted; its invisibility is managed by adjusting the weight of image regularization loss. The weights $w_{adv}, w_{con1}, w_{con2}, w_{con3}, w_{con4}, w_{reg}$ correspond to the fine-tuning attack methods, attacking IP-Adapter Ye et al. (2023a), Reference-only Team (2023), Magic Animate Xu et al. (2024), Ecomimic Chen et al. (2024), and image regularization, respectively. In ANE training, the weights are set to 30, 50, 60, 30, 30, and 200, respectively; in the PGD method, the weights are 3, 5, 5, 2, 2, and 0, respectively.

We implement the Anti-Reference (PGD) method under the following parameter settings. The step size $\alpha$ is set to $1 \times 10^{-3}$, and the number of iterations $T$ is 300. The perturbation is constrained within an $\ell_\infty$ norm ball of 0.05, corresponding to a maximum perturbation magnitude of $\frac{13}{255}$ per pixel. These settings are chosen to balance the attack's effectiveness and noise invisibility. For more implementation details, please see the supplementary materials.

**Baseline methods.** We use PhotoGuard (Salman et al., 2023), AdvDM (Liang et al., 2023), and SimAC (Wang et al., 2024a) as baselines, with SimAC being an improved version of the classic Anti-DreamBooth (Van Le et al., 2023). We systematically evaluate the protection effectiveness of our method and the baseline methods across seven customization generation tasks, including three fine-tuning-based methods: DreamBooth, LoRA, and Textual Inversion; two tuning-free methods: IP-Adapter and reference-only; and two tasks involving human figure animation: Magic Animate and Ecomimic.

**Evaluation benchmarks.** In constructing the evaluation dataset, we follow previous works. For subject-driven generation, we select 10 subject categories from the DreamBooth dataset Ruiz et al.

(2023), with 3 to 5 images per category. For face-driven tasks, we use 10 identities from the CelebA-HQ dataset. For each subject or individual, we generate a total of 200 images using 10 different prompts for quantitative evaluation. For face-driven and body animation tasks, we generate 200 images using CelebA-HQ and TikTok data, respectively, for quantitative comparison.

**Evaluation metrics.** In our evaluation of person-centric image generation quality, we utilized ISM (Identity Score Matching) metrics (Van Le et al., 2023) to assess protection effectiveness, where lower ISM scores indicate more effective disruption of individual identity in the generated images. Additionally, we measured general image quality using Aesthetics Score (AI, 2023) and CLIP-IQA (CLIP Image Quality Assessment) (Wang et al., 2023), which evaluate the naturalness and perceptual quality of images. These metrics were applied across all frames for tasks involving human body and face-driven content. Lower values in these metrics indicate better image protection effectiveness.

## 5.2 QUANTITATIVE EVALUATION

In this section, we present the quantitative evaluation results and time cost for our method and baselines across seven customized generation methods. For all baseline methods, we use their default code and settings to learn adversarial noise. The results of our two methods used for calculating quantitative metrics are all obtained through joint optimization while results of other baselines are optimized independently on each generation method.

**Critical Oversight.** When training Dreambooth with adversarial images, we followed the common practice of not fine-tuning the CLIP text encoder. The protection performance of Anti-Dreambooth and SimAC relies on the flawed assumption that Bob fine-tunes the CLIP text encoder. See the supplementary materials for details.

Table 1: Quantitative comparison on **ISM Score**. Bold values denote best performance.

| Method | Ours (PGD) | Ours (ANE) | Sim AC | Adv DM | Photo Guard | Clean |
|---|---|---|---|---|---|---|
| Dreambooth | **0.029** | 0.078 | 0.051 | 0.077 | 0.081 | 0.287 |
| LoRA | **0.005** | 0.017 | 0.008 | 0.015 | 0.022 | 0.085 |
| Textual Inversion | **0.011** | 0.123 | 0.018 | 0.018 | 0.304 | 0.336 |
| IP-Adapter | **0.197** | 0.226 | 0.225 | 0.225 | 0.242 | 0.233 |
| Reference-only | **0.038** | 0.198 | 0.096 | 0.096 | 0.295 | 0.348 |
| Echomimic | 0.655 | **0.574** | 0.673 | 0.668 | 0.677 | 0.715 |
| Magic Animate | 0.163 | 0.221 | 0.236 | 0.236 | **0.134** | 0.308 |

Table 2: Quantitative comparison on **Aesthetic Score**. Bold values denote best performance.

| Method | Ours (PGD) | Ours (ANE) | Sim AC | Adv DM | Photo Guard | Clean |
|---|---|---|---|---|---|---|
| Dreambooth | **5.345** | 5.716 | 5.687 | 5.874 | 5.935 | 5.985 |
| LoRA | **5.511** | 5.694 | 5.719 | 5.823 | 5.856 | 5.951 |
| Textual Inversion | **4.344** | 4.988 | 4.552 | 5.400 | 5.723 | 5.971 |
| IP-Adapter | **5.548** | 5.930 | 5.771 | 6.050 | 5.961 | 6.241 |
| Reference-only | **4.836** | 5.480 | 4.847 | 5.384 | 5.996 | 6.216 |
| Echomimic | 5.506 | **5.370** | 5.377 | 5.631 | 5.461 | 5.817 |
| Magic Animate | **4.451** | 4.716 | 5.057 | 4.988 | 4.582 | 4.951 |

Table 3: Quantitative comparison on **CLIP-IQA**. Bold metrics represent methods that rank 1st.

| Method | Ours (PGD) | Ours (ANE) | Sim AC | Adv DM | Photo Guard | Clean |
|---|---|---|---|---|---|---|
| Dreambooth | **0.550** | 0.552 | 0.561 | 0.631 | 0.623 | 0.648 |
| LoRA | **0.566** | 0.579 | 0.591 | 0.662 | 0.634 | 0.642 |
| Textual Inversion | **0.444** | 0.462 | 0.500 | 0.599 | 0.583 | 0.653 |
| IP-Adapter | 0.445 | 0.517 | 0.483 | 0.566 | **0.416** | 0.545 |
| Reference-only | 0.584 | 0.608 | **0.341** | 0.523 | 0.473 | 0.622 |
| Echomimic | 0.419 | 0.527 | **0.319** | 0.573 | 0.500 | 0.556 |
| Magic Animate | 0.225 | 0.202 | **0.184** | 0.191 | 0.196 | 0.217 |

Table 4: Time Cost of Defense Methods. Our method (ANE) shows a significant advantage.

| Method | GPU(s) | CPU(s) |
|---|---|---|
| Ours(PGD) | 846 | - |
| Ours(ANE) | 0.21 | 1.05 |
| AdvDM | 212 | - |
| PhotoGuard | 66 | - |
| SimAC | 51 | - |

Table 5: Our method matches SOTA performance in adversarial noise invisibility.

| Method | PSNR (↑) | SSIM (↑) |
|---|---|---|
| Ours(PGD) | 30.39 | 0.762 |
| Ours(ANE) | 29.00 | 0.713 |
| AdvDM | 38.04 | 0.939 |
| PhotoGuard | 32.25 | 0.822 |
| SimAC | 32.17 | 0.811 |

**Effectiveness.** From Fig. 3 and Tables 1 to 3, it is evident that our two methods exhibit more comprehensive and thorough attack effects compared to the baseline. Our PGD method effectively protects images from the threats of 7 customized generation methods, and our ANE method also demonstrates effectiveness across all tasks. Specifically, in terms of the most critical ISM metric for measuring the effectiveness of attacks, our method achieved leading results. Our method also holds certain advantages in the Aesthetic-Score and CLIP-IQA metrics.

**Time Cost.** Table 4 shows a comparison of the time required to protect a single image using our method versus the baseline methods. Our method takes only one thousandth of the time required by the baseline methods. This improvement in efficiency marks a crucial advancement from academic research to practical application, laying the foundation for real-world implementation in AI security.

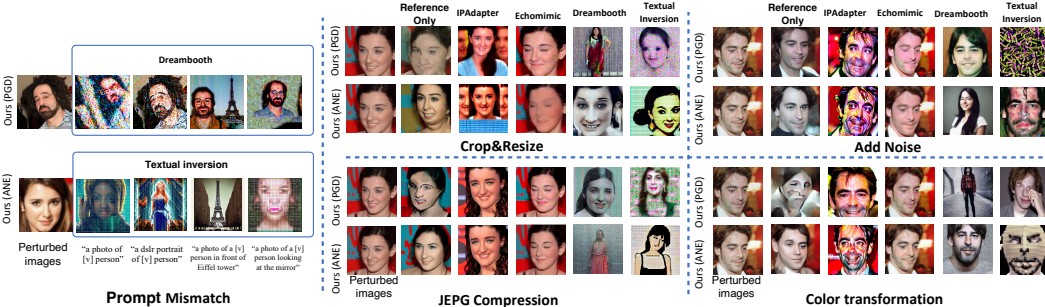

Figure 4: Qualitative Evaluation of Method Robustness. Our method is Robustness under prompt mismatch and image transformation.

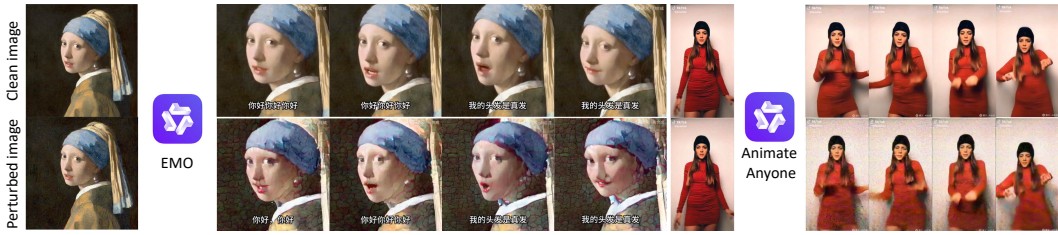

Figure 5: Gray-box Attack on Tongyi APIs. Our method successfully compromises both face- and body-driven generation.

**Invisibility.** Table 5 shows a comparison of adversarial noise invisibility. Compared to the baseline, our method produces slightly more noticeable noise, with a trade-off between invisibility and effectiveness. Our approach, which attacks multiple customized generation methods, faces greater convergence challenges than single-task methods, making comparable invisibility difficult to achieve.

## 5.3 GRAY-BOX PERFORMANCE

In this section, we demonstrate the gray-box transferability of our method. We tested the closed-source face-driven method EMO (Tian et al., 2024) and body-driven method Animate Anyone (Hu, 2024) on the Tongyi app ton (2024). Without access to model parameter, our method shows excellent gray-box transferability, with noticeable artifacts in their outputs.

## 5.4 ROBUSTNESS TEST

**Prompt Mismatch.** When Bob customizes concepts with Stable Diffusion, his prompts may differ from Alice's assumptions during noise generation. PGD-based methods (Van Le et al., 2023), typically trained with fixed prompts (e.g., "a photo of sks person"), suffer under prompt shifts. As shown in Fig. 4, ANE trained on large-scale image-text pairs remains robust to such mismatches.

**Image Transformations.** Our method is robust to common image transformations, such as JPEG compression, crop & resize, noise addition, and color transformations. See supplementary materials for more quantitative results. Our method demonstrates significantly stronger robustness compared to baseline approaches.

## 6 CONCLUSION

This paper introduces Anti-Reference, a novel and effective method for protecting images from the threats posed by mainstream Reference-based generation methods and fine-tuning-based methods. Utilizing a Noise Encoder based on the DiT architecture and a unified loss function, our approach offers universal and efficient protection against various adversarial attacks. Additionally, the introduction of data augmentation techniques and black-box transfer capabilities through white-box proxy models ensures robust and scalable defenses. Extensive experiments validate the effectiveness of Anti-Reference in protecting images from unauthorized customized generation, setting a new standard in the fields of privacy protection and information security.

## CODE OF ETHICS

The authors have read and acknowledge adherence to the ICLR Code of Ethics.

## ETHICS STATEMENT

All datasets used in this work are publicly available and widely adopted in the research community. We comply with dataset licenses and usage guidelines. Human figures appear only as part of these existing benchmarks to evaluate generalization across diverse visual domains. No private or newly collected human data was used.

## REPRODUCIBILITY STATEMENT.

All datasets, model configurations, and training details used in this work are described in the paper. We will release the synthetic paired human–robot dataset, model checkpoints, and inference scripts upon publication to facilitate full reproducibility. Hyperparameters, architecture details, and evaluation metrics are explicitly documented. We also provide ablation studies to clarify the effect of each component. Together, these measures ensure that researchers can replicate and extend our results without ambiguity.

## USE OF LARGE LANGUAGE MODELS

We only used large language models such as GPT-4 and GPT-5 to assist with English grammar refinement and error correction at the writing stage. All technical content—including method design, experimental setup, and quantitative results—was independently conceived, implemented, and verified by the authors. Large language models were not used to modify any experimental data or code. This guarantees the scientific integrity and originality of this work.

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

Supplementary Materials of "Anti-Reference: Universal and Immediate Defense Against Reference-Based Generation"

## A    Critical Oversight

It is worth noting that when training Dreambooth with adversarial images, we did not fine-tune the CLIP text encoder, which aligns with the common practice in the community. We found that the good protection performance of Anti-Dreambooth and SimAC is based on the incorrect assumption that Bob will fine-tune the CLIP text encoder. As shown in Figure 6, when Bob does not fine-tune the CLIP text encoder during Dreambooth training, both of these image protection methods show a significant drop in performance, regardless of whether the CLIP text encoder was fine-tuned during the noise learning process. Our method does not suffer from this issue.

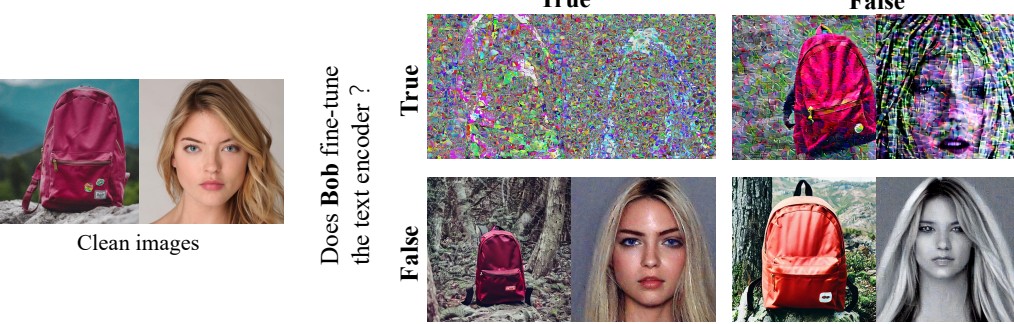

Figure 6: We have identified a critical oversight in the current SimAC method; when Bob does not train the Text Encoder while training Dreambooth, the protection effectiveness of the images is significantly compromised.

## B    Detail of Evaluation Metrics

For evaluating the quality of person-centric image generation, we used widely adopted metrics ISM (Van Le et al., 2023) to quantify the generation quality, where lower ISM represent better protection effectiveness. Additionally, we employed two general image quality assessment metrics, Aesthetic Score (AI, 2023) and CLIP-IQA (Wang et al., 2023). For human body and face-driven tasks, we calculated quantitative metrics across all frames.

- **ISM** (Identity Score Matching): Measures the cosine similarity between the features of the generated face and the original face to evaluate how well the generated image maintains the identity of the subject.
- **Aesthetic Score**: An aesthetic assessment metric that utilizes a linear estimator built on top of CLIP to predict the aesthetic quality of images.
- **CLIP-IQA** (CLIP Image Quality Assessment): Uses CLIP (Contrastive Language-Image Pretraining) to evaluate the perceptual quality of images by assessing how well the visual features of the image align with text descriptions.

## C    Transferability of Adversarial Noise Across Model Architectures

Due to the architectural differences among SD1.5, SD2.0, and SD-XL, their latent spaces significantly differ. We have conducted experiments with adversarial noise on SD1.5, but it could not be generalized to SD-XL. This issue is not unique to our method; there are no successful transfer precedents in this field. Table. 6 shows the transferability results for Anti-Reference, where noise can be transferred

between SD1.4 and SD1.5 due to their similar architectures and latent spaces. We perform a joint attack across models with different architectures, and experimental results show that this strategy effectively enables simultaneous attacks on methods with varying backbones.

Table 6: Transferability results of adversarial attacks across different SD architectures.

| Attack | SD1.4 | SD1.5 | SD2.0 | SD-XL |
|---|---|---|---|---|
| SD1.5 | ✓ | ✓ | ✗ | ✗ |
| SD2.0 | ✗ | ✗ | ✓ | ✗ |
| SD-XL | ✗ | ✗ | ✗ | ✓ |
| Joint attack on SD1.5, 2.0, XL | ✓ | ✓ | ✓ | ✓ |

## D  HUMAN EVALUATION

To further validate the effectiveness of our proposed methods in perceptual scenarios, we conducted a human evaluation study via an online questionnaire. Participants were presented with a series of images generated by different models using both clean and adversarial inputs. They were asked to determine whether each image exhibited visible artifacts or distortions. All images were presented in randomized order, and participants were not informed which ones contained adversarial perturbations to minimize bias.

A total of 30 participants took part in the evaluation, each reviewing 50 image samples. For each image, they were instructed to answer two questions: (1) whether the image contained visible artifacts, and (2) whether it exhibited noticeable distortions. The evaluated samples included adversarial images generated by our two proposed methods: PGD (Projected Gradient Descent) and ANE (Adversarial Noise Embedding).

As shown in Tab. 7, the results demonstrate that the PGD method is highly effective at introducing perceptible artifacts. Meanwhile, our ANE method also achieves strong perceptual impact, producing noticeable distortions in the generated images. Both methods successfully mislead the generation models while being perceptible to human observers, highlighting their practical utility and robustness in adversarial attack settings.

Table 7: Percentage of users judging the attack as successful (obvious artifacts observed). Bold metrics indicate top-ranked methods.

| Method | Ours(PGD) | Ours(ANE) | SimAC | AdvDM | PhotoGuard |
|---|---|---|---|---|---|
| Dreambooth | **100** | **100** | 93 | 85 | 90 |
| LoRA | **97** | 92 | 94 | 69 | 93 |
| Textual Inversion | **100** | **100** | 96 | 85 | 89 |
| IP-Adapter | **93** | 89 | 72 | 65 | 67 |
| Reference-only | **96** | 94 | **96** | 87 | 93 |
| Echomimic | **100** | **100** | **100** | 94 | 98 |
| Magic Animate | **100** | **100** | **100** | **100** | **100** |

## E  LIMITATIONS AND FUTURE WORK

While our method inherits the common challenge of imperceptible adversarial cues—shared by most SOTA defenses—it remains effective in disrupting generation outputs across models. Our approach is built on SD 1.5 to align with widely-used reference-based generation systems, with results on SD-XL and SD3 included in the supplementary. Future work will extend compatibility to emerging architectures such as Diffusion Transformers.

## F   MORE ROBUSTNESS TEST RESULTS

Figure 7 and 8 shows that our method is robust to common image transformations, such as JPEG compression, crop & resize, noise addition, and color transformations.

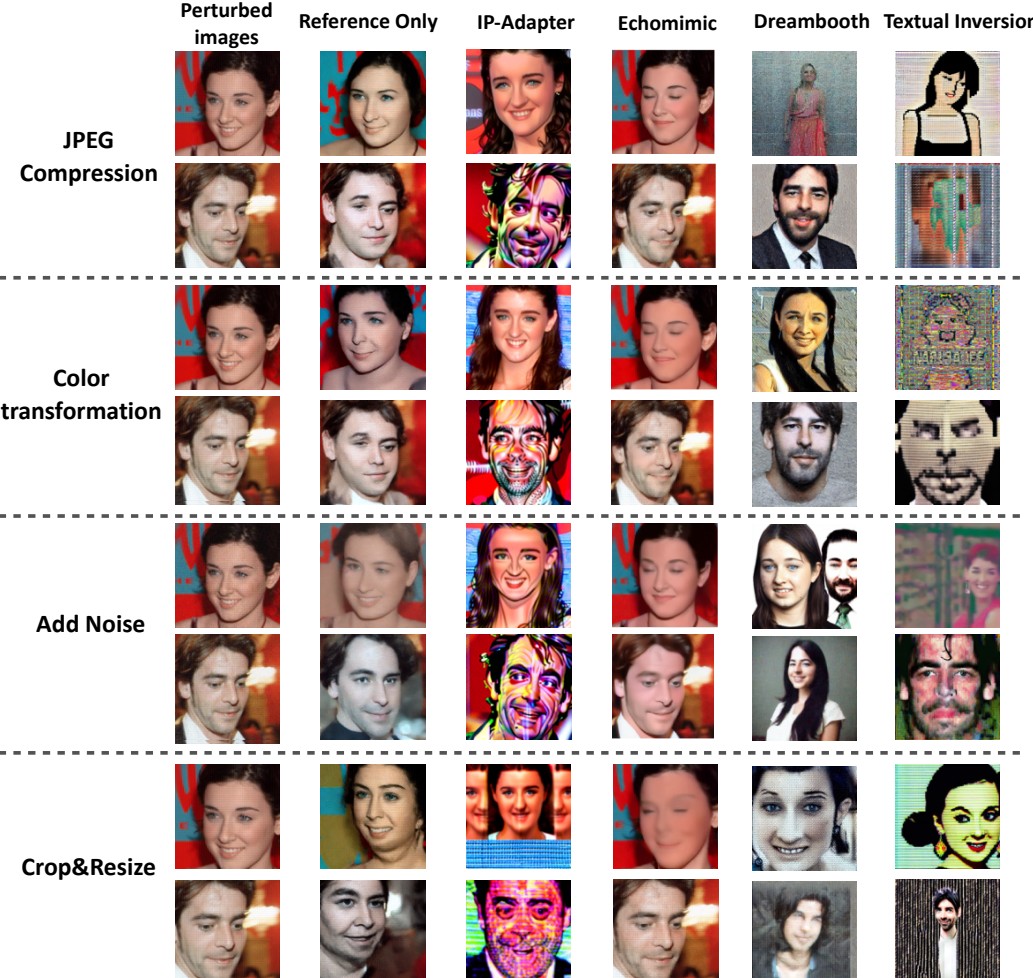

Figure 7: More robustness test results: Our method (ANE) is robust against common image transformations.

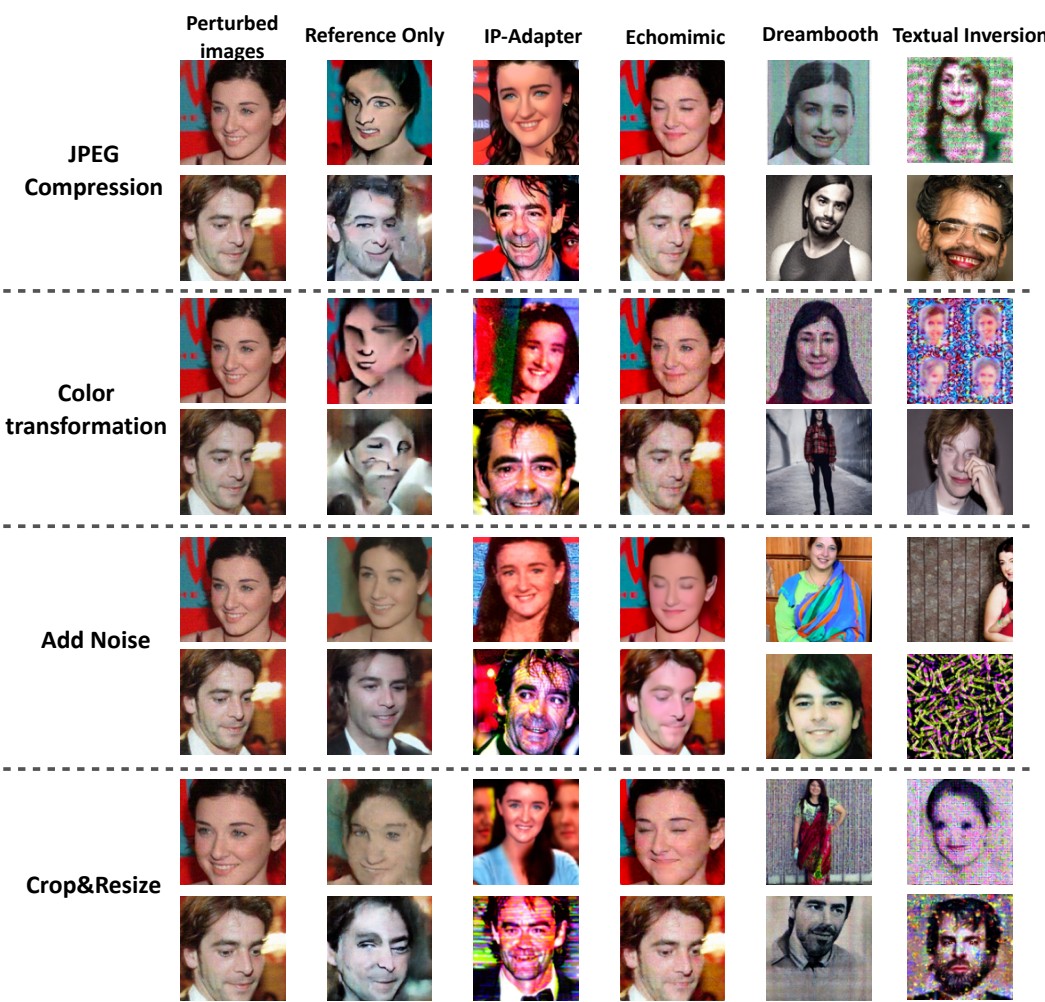

Figure 8: More robustness test results: Our method (PGD) is robust against common image transformations.

