# OpenReview forum: "Anti-Reference: Universal and Immediate Defense Against Reference-Based Generation"
_ICLR.cc/2026/Conference — ICLR 2026 Conference Withdrawn Submission_

### Official Review · Reviewer_F22m · 2025-10-20

**Soundness:** 3
**Presentation:** 2
**Contribution:** 3
**Rating:** 2
**Confidence:** 4

**Summary:**

The paper introduces Anti-Reference, a universal image protection framework designed to defend against both fine-tuning-based (e.g., DreamBooth, LoRA) and reference-based (e.g., IP-Adapter, Magic Animate, EMO) generative models. The method perturbs user images with imperceptible adversarial noise, either by training a DiT-based Adversarial Noise Encoder (ANE) or by optimizing noise directly via PGD. The unified loss jointly disrupts diffusion and conditional generation processes, while data augmentations enhance robustness to real-world transformations. Additionally, the authors explore gray-box transferability by attacking proxy models with architectures resembling proprietary APIs.

**Strengths:**

1. The paper is the first to explicitly target reference-based generation methods (e.g., IP-Adapter, Magic Animate, EMO), while simultaneously offering a universal perturbation framework that generalizes across both fine-tuning-based and non-finetuning generative pipelines.
2. The paper conducts quantitative and qualitative experiments across 7 customized generation tasks, evaluates robustness to prompt and transformation variations, and provides human evaluation studies - showing methodological rigor and practical reliability.

**Weaknesses:**

1. As shown in Tables 1-3, the ANE variant consistently underperforms compared to baselines such as SimAC, AdvDM, and PhotoGuard across most target generative models. For example, under the ISM Score metric (Table 1), ANE lags behind baselines in both fine-tuning-based and reference-based settings. Also, while the paper highlights that ANE runs ~1000× faster than baseline methods, its lower protection performance raises doubts about the practical utility of such speed. Conversely, the PGD variant achieves stronger performance but incurs at least 4× higher computational cost than the baselines. This trade-off is not fully addressed or analyzed, leaving the core value of ANE somewhat unclear.
2. The paper evaluates performance using ISM, Aesthetic Score, and CLIP-IQA. However, prior works like PhotoGuard and AdvDM typically report FID and Precision, which are more widely adopted and allow for better comparability. The use of unconventional metrics hinders direct evaluation against existing approaches.
3. The comparison in Table 5 shows that the proposed methods operate under different PSNR and SSIM levels than baselines (e.g., PSNR of the proposed method is about 30 and that of  AdvDM is about 38). Without equalizing the perturbation strength, it's difficult to fairly assess whether improvements stem from better framework design or simply from allowing more visible (and thus stronger) noise.

Note: Weaknesses 1-3 correspond directly to Questions 1-3.

**Questions:**

1. What is the practical justification for introducing ANE, given that its protection performance is consistently lower than both the baselines and your own PGD variant? Does the runtime gain outweigh the significant drop in protection quality?
2. Can the authors report results using widely adopted evaluation metrics, such as FID and Precision, to enable more direct comparison with prior works?
3. Can the authors report additional experiments where PSNR/SSIM are matched across methods, to ensure the perturbation budgets are comparable? For instance, increasing ANE's regularization weight could improve its perceptual quality. Were such variants tested?
4. In Figure 5 (EMO, 5th column), it is unclear whether the clean and perturbed outputs are meaningfully different. Also, if the protection effect relies on generating visible artifacts, couldn’t an attacker simply denoise the generated image? As universality is a central claim, quantitative gray-box results would strengthen the argument.
5. According to Section 5.1, the unified loss employs fixed weight values. Could the authors clarify the rationale behind these choices and provide ablation studies to validate their impact on performance?

---

### Official Review · Reviewer_ySbQ · 2025-10-26

**Soundness:** 2
**Presentation:** 3
**Contribution:** 2
**Rating:** 2
**Confidence:** 4

**Summary:**

This paper introduces Anti-Reference, a universal defense framework designed to protect personal images from misuse in reference-based diffusion models. The authors propose an Adversarial Noise Encoder (ANE) that efficiently generates protective perturbations in a single forward pass, offering much faster inference than PGD-based methods, though with somewhat weaker protection performance. Extensive experiments across multiple reference-based generation tasks demonstrate that Anti-Reference effectively degrades identity fidelity and visual consistency.

**Strengths:**

- This paper introduce a universal and model-agnostic defense framework applicable to DreamBooth, IP-Adapter, MagicAnimate, and Ecomimic without per-model retraining.
- The authors propose an Adversarial Noise Encoder (ANE), a ViT-based module that generates protective noise in a single forward pass, which is faster than PGD and demonstrates strong cross-model transferability by effectively performing gray-box attacks on unseen models.
- The experiments provide comprehensive quantitative and qualitative evaluations across seven reference-based generation tasks and several datasets.

**Weaknesses:**

- The paper presents an empirical approach with limited theoretical justification.
- The authors claim a universal framework but only validate on human-centric datasets, limiting generalizability.
- Section 4.3 lacks depth and insight, and it reads like a simple combination of loss terms without analysis behind the design choices.
- No ablation study on the contribution or sensitivity of each loss component.
- No failure case analysis to understand when and why the protection fails.
- The gray-box evaluation (Section 5.3) is only supported by qualitative examples (Figure 5) without quantitative metrics.
- Baseline configurations are under-specified; it is unclear whether comparisons are conducted under fair and consistent settings.

### Minor
- Figure 5 is not referenced or explained in the main text.
- The formatting of Table (Table x / Tab. x) and Figure (Figure x / Fig. x) references should be unified for consistency.

**Questions:**

- The inclusion of a human evaluation is commendable; could the authors provide statistical significance tests to strengthen the conclusions?
- Is the proposed defense easily bypassed or neutralized through **fine-tuning** of the target model?
- In Section 5.1, could the authors explain the rationale behind their chosen experimental parameter settings?
- Could the authors provide training curves for the total loss to illustrate how each component evolves during training and to better demonstrate the optimization stability and convergence behavior?
- Other questions are discussed in the **Weaknesses** section.

---

### Official Review · Reviewer_P5V7 · 2025-10-27

**Soundness:** 3
**Presentation:** 3
**Contribution:** 3
**Rating:** 4
**Confidence:** 4

**Summary:**

This paper proposes Anti-Reference, a new defense method designed to protect user images from being exploited by reference-based generation techniques in diffusion models (e.g., IP-Adapter, ReferenceNet, DreamBooth, LoRA). Such methods are widely used for customized image and video generation, including talking-face and human animation, but can also be abused to create harmful or misleading content such as deepfakes, fake news, or explicit imagery.

**Strengths:**

1. The paper addresses an important and emerging issue of defending against misuse of diffusion models for reference-based generation, which is socially relevant.

2. he method attempts to unify defenses across fine-tuning, non-fine-tuning, and human animation methods.

3. The Adversarial Noise Encoder (ANE) is designed to be much faster than PGD-based baselines, which is practically attractive.

**Weaknesses:**

1. The paper frames its method as a universal defense, yet all experiments are constrained to Stable Diffusion 1.5 and related architectures. Transferability to SD-XL or newer backbones is minimal, undermining the universality claim. Attacks on commercial APIs are anecdotal and lack quantitative rigor, making the “gray-box” transferability evidence weak. The evaluation is narrowly scoped to curated datasets (LAION, CelebA, TikTok) and does not convincingly prove robustness in real-world, diverse image-sharing platforms.

2. The technical core of the paper is simply adding adversarial noise through a Vision Transformer–based encoder and combining existing loss terms with weighted sums. This is more of an engineering adjustment than a conceptual innovation. The “unified loss” function is not theoretically justified and provides little new understanding of why the method should generalize. Without rigorous analysis or principled novelty, the contribution feels incremental, essentially presenting an efficiency-oriented version of existing adversarial perturbation work.

3. The choice of baselines is questionable, and comparisons are not entirely fair. For example, Anti-DreamBooth is dismissed on the grounds of CLIP encoder fine-tuning, but no robust alternative baseline is included. Metrics such as ISM, Aesthetic Score, and CLIP-IQA are convenient but insufficient to capture real-world usability or perceptual quality. Human evaluation is limited to a small, non-diverse group, undermining the strength of user-centered claims. Time cost comparisons are also overstated: PGD is portrayed as impractically slow, yet the authors ignore optimized adversarial methods that could serve as stronger baselines.

4. The method concedes that its noise introduces more visible artifacts than baselines but downplays this critical usability flaw. For real users, any perceptible distortion undermines adoption, making the defense questionable in practical contexts. Moreover, the robustness tests are limited to trivial augmentations like JPEG compression and cropping, leaving open whether the defense would survive complex transformations applied by real platforms. Finally, the ethical discussion is superficial, consisting of boilerplate acknowledgments without engaging with the broader societal consequences of deploying adversarial noise protections at scale.

**Questions:**

I will raise my score if the authors address W3 and W4.

---

### Official Review · Reviewer_JeS9 · 2025-10-30

**Soundness:** 3
**Presentation:** 3
**Contribution:** 2
**Rating:** 4
**Confidence:** 3

**Summary:**

This paper presents Anti-Reference, a defense mechanism that protects personal or artistic images from being exploited in reference-based or fine-tuning-based diffusion model generation. The paper claims strong performance, transferability to gray-box commercial APIs, and orders-of-magnitude faster inference compared to PGD-based methods.

**Strengths:**

1. The paper tackles a relevant problem of defending images from misuse in reference-based generation by using an adversarial perturbation framework that is reasonably well-engineered and empirically validated.
2. the paper is well written and well presented

**Weaknesses:**

1. The main empirical results are reported on Stable Diffusion 1.5 (SD 1.5), which is increasingly outdated in the context of generative models. Given that SD 3 and other more advanced architectures are now available and widely used.  Without experiments on these newer models, it is uncertain whether the proposed methods generalized to current-state generation systems.
2. Gray-box setting: The gray-box setting used in this paper is only partially convincing. While it is reasonable from a practical standpoint, the experiments mainly involve models that share the same SD 1.5 backbone, making the evaluation closer to an intra-family transfer rather than a true gray-box or black-box setting. Consequently, the claimed “universality” is overstated.
3. Limited technical novelty: While the paper presents an interesting unification of adversarial defense mechanisms against both fine-tuning–based and reference-based diffusion models, the underlying techniques rely heavily on established adversarial training principles. The primary novelty lies in combining these components into a single framework rather than introducing a fundamentally new algorithmic insight.
4. Lack of baselines:  The comparison set of defense baselines is incomplete. While the paper includes several prior works, it omits other recent or conceptually related defenses such as data-free universal watermarking methods.

**Questions:**

See above

---

### Note · Authors · 2026-01-19

I have read and agree with the venue's withdrawal policy on behalf of myself and my co-authors.